# Concept for Digital Product Twins in Battery Cell Production

Achim Kampker, Heiner Hans Heimes, Benjamin Dorn, Henning Clever *[ID], Robert Ludwigs *[ID], Ruiyan Li [ID] and Marcel Drescher

Production Engineering of E-Mobility Components (PEM), RWTH Aachen University, 52072 Aachen, Germany; b.dorn@pem.rwth-aachen.de (B.D.)

*   Correspondence: h.clever@pem.rwth-aachen.de (H.C.); r.ludwigs@pem.rwth-aachen.de (R.L.)

**Abstract:** This paper presents an approach for the design and derivation for establishing a digital product twin for battery cells. A digital product twin is a virtual replica of a physical battery cell and can be used to predict and optimize quality properties and performance in real-time. The study focuses on pouch cell manufacturing and aims to map the large amount and variety of process information down to purchased parts and interim products. The approach for this study was to collect and analyze data from the physical production process and use this information to structure a digital battery product twin based on its product architecture. The main findings of this study indicate that a digital product twin can be effectively structured and implemented in a digital interface based on its product architecture in combination with data from the physical production process. The results of this study show the potential of digital product twins, in which statements about material, design, and behavior can be made using real information from production. Further research will focus on the practical application and implementation of digital product twins in a battery cell pilot production.

**Keywords:** battery cell production; pouch cell; digital product twin; digitalization; industry 4.0

## 1. Introduction

The storage and use of renewable energy sources is a key factor in the automotive industry for becoming independent of fossil fuels and minimizing greenhouse gas emissions, leading to a rapid growth of electric mobility and an enormous increase in the demand for high-performance and sustainable battery cells in the automotive industry. Several announcements have been made by OEMs and battery cell manufacturers, especially in Europe, to meet the global demand for battery cells. Global demand for lithium-ion batteries is expected to exceed 4 TWh in 2030, with planned battery factories in Europe covering about one-third of the global market [1,2].

Battery cell production is a complex process that involves multiple stages, including design, manufacturing, and quality control. To ensure high-quality and consistent output, it is essential to have a complete understanding of both the product and the production process. This information can be used to identify areas for improvement, optimize production processes, and minimize waste [3,4]. The use of a digital twin can facilitate this by providing a comprehensive view of the product and process, enabling real-time monitoring and analysis. The basis for these approaches is the data framework on which the digital twin is built. This presents a challenge in that the data generated are both voluminous and heterogeneous. Battery cell production entails a highly complex process chain, consisting mostly of material-property-altering processes, which are performed in batches, quasi-continuously, as well as discretely. For this reason, there is a need for mechanisms that ensure the interoperability of all product and process data [5,6]. Moreover, current publications on digital twins in battery production do not provide in-depth information on which data are relevant across the process chain and how to structure them.

The process of digitalization involves converting analog product characteristics into digital form, which facilitates the electronic and informational transfer, storage, and pro-

cessing of data. The objectives for the digitalization of battery cell production are ambitious. The goals range from fully automated decision-making to the adjustment of parameters by artificial intelligence for zero-scrap production, resulting in networked and intelligent battery production facilities from raw material to finished cell [7]. Digital twins are digital representations of physical objects or systems. The idea of digital twins was first proposed by NASA in the context of its Apollo program as a way to test problems and solutions related to a physical object, such as spacecraft. Over time, the physical object was replaced with a digital image or simulation. The concept of digital twins offers a new way of thinking about physical objects and systems, providing a digital representation that can be used to improve efficiency and innovation [8].

A digital twin refers to a virtual copy of a physical system that can be used for performance optimization and behavior prediction. Any type of simulation and modeling is an application of this digital twin and focuses on specific aspects, for example material behavior, fluid dynamics, or thermal performance. However, creating a digital twin requires the integration of all available information about the system, including real-time data, historical data, and both process and material data. This integration is a complex task, and the importance of the interface structure of a digital twin cannot be overstated.

Current approaches, specifically in battery cell production, present a digital product twin either solely as a generic view for universal data capturing or as a selected simulation model of a specific feature (see Section 2). There is currently no common understanding of how the data structure for a digital twin of a battery cell needs to be organized in order to leverage a variety of data- and model-based applications. The aim of this work is to investigate and derive a potential concept of digital product twins as an information model in battery cell production in order to unify and integrate all information into a single, coherent digital product twin (see Section 4). In Section 2, the study focuses on the fundamentals of digital twins in a manufacturing context and proposes a methodological approach for deriving a digital product twin. Afterwards, the proposed approach is applied in Section 3 for battery cell production, providing product understanding, process understanding, and mapping the information of product and process. The derived concept for a digital product twin for battery cells is elaborated in Section 4. The key features of the concepts for a digital product twin in battery cell production is the use of the battery cell's product structure in combination with production parameters along the process chain. This is illustrated and functionalized through systematic data structuring. The conclusion of this paper provides an outlook on possible extensions and adaptations of the digital product twin through the integration of field data and includes suggestions on how battery cells can be implemented in pilot line manufacturing.

## 2. Fundamentals and Approach

In the following paragraph, the main aspects of the concept of digital twins in a production context are presented, and the current state of research is carefully discussed. Related work on digital twins and, in particular, recent studies on digital product twins will be reviewed and presented. Based on the findings and identified gaps, a structuring concept for a digital product twin in battery cell production will be derived.

The general concept of digital twins is based on the idea of creating a virtual representation of a physical asset, process, or system in real-time, using data from various sources. This virtual model is then used to simulate the behavior, performance, and interactions of its real-world counterpart. In the manufacturing context, digital twins are said to have the potential to revolutionize the way manufacturing and production processes are managed and optimized. The areas of application for digital twins are vast and range from improving operational efficiency and reducing costs to optimizing production processes and improving product design. For instance, digital twins can be used to simulate an assembly line to test different scenarios to optimize a production process and minimize waste. In a production process, a digital twin can be used to monitor equipment performance, predict potential failures and maintenance requirements, and optimize the production process in

real time [9,10]. Figure 1 illustrates an overview of the general functions and compositions of digital twins.

**Figure 1.** Five-dimension model (left) and composition and application of digital twins (right). Adapted from [9].

In order to gain more insights into existing approaches for establishing digital twins in a production context, the following chapter analyzes related work in this field. In particular, comprehensive discussions on digital twins and the most recent findings will be addressed. A special focus is given to digital product twins.

### 2.1. Related Work on Digital Product Twins

There are various approaches for conceptualizing digital twins in the literature. In the following, recent approaches and methods from the planning of production systems, product engineering, and software development are presented, which can partially be applied to solve the problem presented above.

A structured review for the systematization of the digital twin concept in industry has been conducted by Sjarov et al. (2020). They conclude that many publications actually avoid defining explicitly the concept of a digital twin for themselves and rather implicitly provide a set of abilities and properties associated with a digital twin. As a result, the term digital twin can be paraphrased as a multi-domain simulation, a computerized counterpart of a physical system, a virtual representation of what has been produced, a virtual substitute of real-world objects, an integrated simulation and forecasting tool, or a linked collection of digital artifacts [11]. Another general definition and common understanding of the concept as well as relevant terms of digital twins are presented by Bergs et al. (2020) by highlighting their requirements and showing specific use cases of implementation and application. They generally define that digital twins exist for physical assets, which are real objects or systems that undergo process-related state changes [12]. In the context of circular economy and material flow management, Preut et al. (2021) introduce the term digital twin as a digitization concept to enable relevant product information being available to the right stakeholder at the right time considering information requirements in individual process steps of different stakeholders [13]. Bonney et al. (2022) introduce two main periods of interest regarding digital twins, distinguishing between pre-delivery usage related to design and manufacturing, and asset management for lifecycle determination and verification and validation [14].

In terms of conceptualizing and deriving a digital product twin, the following additional approaches can be found in the literature.

Boschert et al. (2018) describe in general terms that the digital product twin comprises all design artifacts of a product [15]. According to Wu et al. (2020), the data composition of a digital product twin mainly includes product design, data, product service data, and

product retirement and scrap data [16]. Further, Bartelt et al. (2022) describe the digital product twin as the first step for the engineering of a production plant. The purpose should be for the digital product twin to represent the real product as accurately as possible and to contain all data, models, and information obtained, for example, from CAD data, component types, connections, or the dimensions and positions of the products, among other things (e.g., geometric data, schematics, material properties, etc.) [17]. Eickhoff et al. (2022) provide a description of a flexible approach to a digital product twin that focuses on product lifecycle management, and present an approach for the design and implementation of digital twins. In this context, metadata-based integration of product description based on variant-specific BOMs is introduced, creating a corresponding metadata graph node between data objects (i.e., parts and assemblies) [18].

Wagner et al. (2019) outline the integration of a digital product twin of the underlying product, which enables the application of new feature-oriented strategies for quality control in production. Thus, during the development of a digital product twin, by linking a product twin to a production twin, the requirements for later use in production or application can already be directly planned and implemented [19]. Here, Onaji et al. (2022) also introduce an integrated digital product process twin that uses the product twin to influence the process configuration. The product twin, which is based on the product specifications, can influence the configuration of the production system through the digital process twin. On the other hand, the production system can provide the product data through the digital process twin to represent the physical product [20].

More generally, Göbel et al. (2020) state that the design of digital product twins should not be an end in itself. Rather, it must be designed in view of immediate application objectives or a strategically defined, future range of applications. Such objectives may include application scope, process orientation, product orientation, and lifecycle integration. An exemplary view of the digital product twin is displayed to a service technician through the bill of materials structure, i.e., assemblies and components of a tractor land machine [21]. This is demonstrated by Prior et al. (2022), who focus on developing a metamodel of a digital product twin and providing a template for the necessary data framework in AutomationML. For this purpose, AutomationML class libraries (such as attribute types, role classes, extended information, attribute assignment, etc.) were created to describe the digital twin [22]. Meanwhile, Zheng et al. (2022) address the integrated modeling of large dynamic datasets as an important task to be solved for digital product twins. In this context, the goal of data modeling is determined by analyzing the properties of different data types for the digital product twin, and the properties of different data storage modes are investigated. The structure, attributes, and scale properties of dynamic product data are used to improve database performance [23].

For battery cell production, Kies et al. (2022) describe that quality prediction during the production process can add significant value for all parties involved in production. However, the concept presented superficially indicates that properties, process parameters, and quality metrics are given as input data for the digital twin of the product [24]. Krauß et al. (2023) suggest that a digital product twin contains information, quality data, and other characteristics about the raw materials and all intermediate and final products, including the parameters of the various processing operations. In the case of battery cells, this includes the electrode pastes and electrode rolls for the anode and cathode, which are combined in the digital twin of the battery [2]. Yet, an approach for future implementation is not conceptualized or outlined.

In summary, it can be seen that some concepts for a digital twin and in particular also a digital product twin have already been postulated. However, all authors describe the higher-level concept in terms of individual product families. In addition, reference is made to the challenges that the successful implementation of digital product twin requires a traceability system that semantically and contextually assigns the accruing data. For battery cells, there is still a lack of a concept to link different types of information from the various intermediate products.

*2.2. Methodological Approach*

Based on the review of related work regarding digital product twins, a methodological approach for deriving a concept for digital product twins in battery cell production is proposed. Figure 2 shows the general steps comprising product understanding, process understanding, information mapping, and digital twin design.

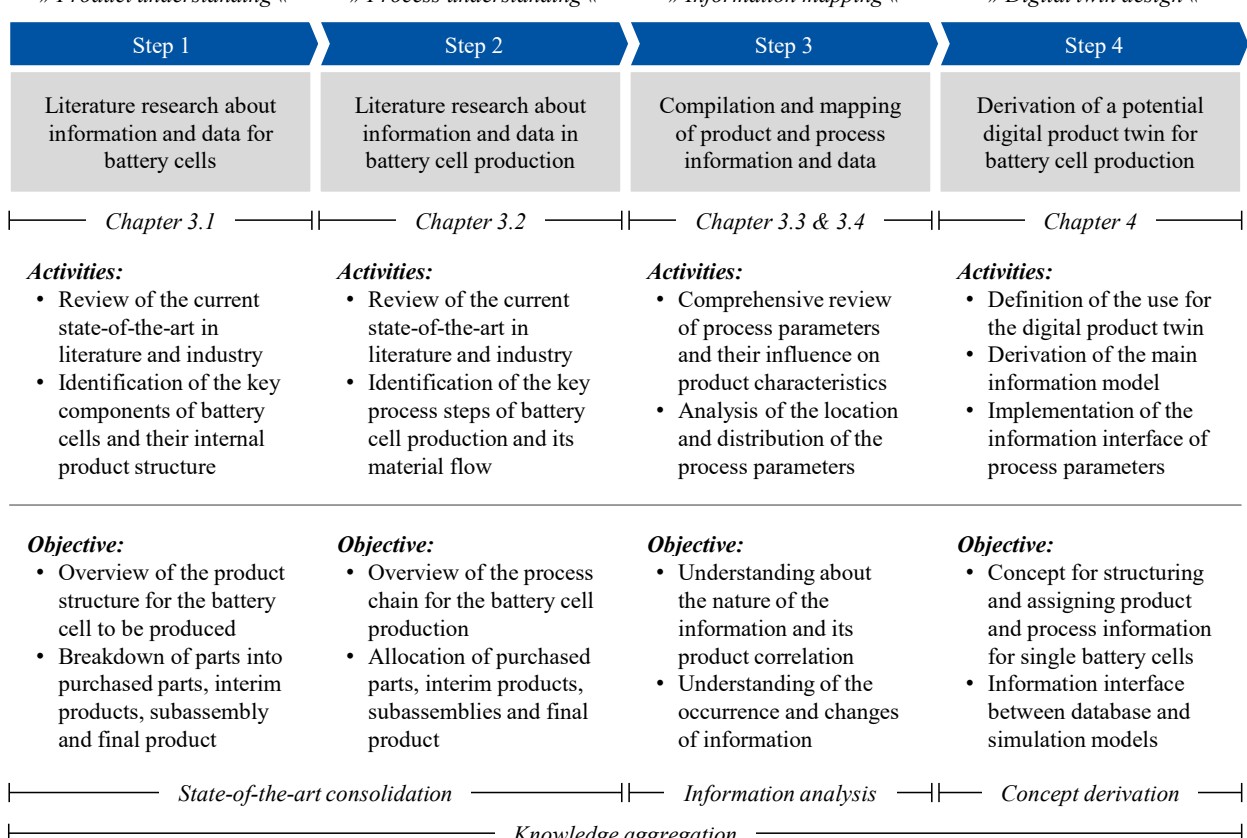

**Figure 2.** Approach for the derivation of a digital product twin in battery cell production.

In order to derive a digital product twin, a general product understanding is essential. Therefore, the first step begins with an extensive literature research about information and data for battery cells and its underlying product structure including components, assemblies, purchased parts, etc. Afterwards follows process understanding, where all information and data regarding the process chain and manufacturing steps in industrial battery cell production are considered. This includes a detailed understanding about any purchased parts and interim products along the process chain. During information mapping, the gathered product and process information is compiled and allocated to build the foundation for a concept of a digital product twin for battery cells. The final step concludes with the derivation of a potential digital product twin for battery cell production. This includes its implementation and structuring.

All steps of the proposed methodology follow in the next chapters. Product and process understanding are discussed in particular in Section 3, along with contextualization and mapping of the information. The derivation of a concept for a digital product twin for battery cell production is presented in Section 4.

## 3. Product and Process Information in Battery Cell Production

In this chapter, the structure of a lithium-ion battery cell and the process chain for battery cell production are discussed following the methodological approach. This study will be based on a pouch cell. Afterwards, the process and quality parameters for interim

products will be analyzed and categorized, followed by mapping of the product and production information.

### 3.1. Structure of a Lithium-Ion Battery Cell

The general structure and design of a lithium-ion battery is strongly dependent on the underlying cell format. Three cell formats have become established in the automotive industry: round, prismatic, and pouch cells. This paper will elaborate a concept for a digital product twin for lithium-ion batteries based on the pouch cell format. Figure 3 shows the general structure of a pouch cell through its product architecture [25,26].

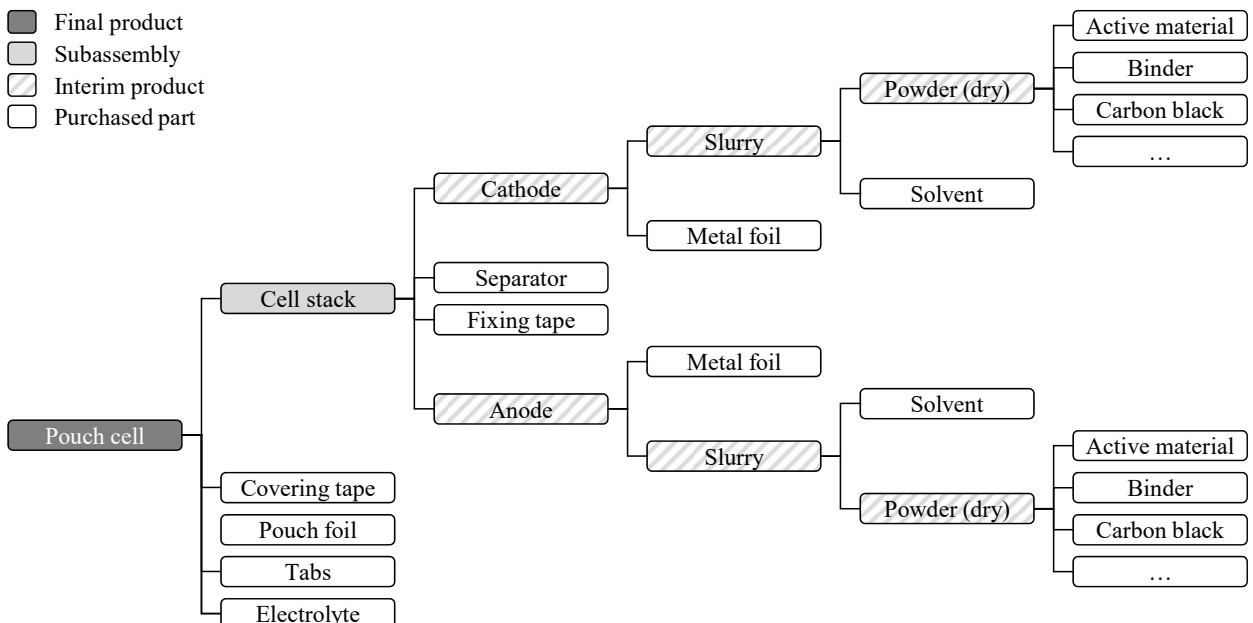

**Figure 3.** Product structure of a lithium-ion pouch battery cell.

A distinction is made between the final product, subassemblies, interim products, and purchased parts. A subassembly here refers to an object consisting of two or more parts, which can generally be disassembled again. In contrast, an interim product refers to partly finished goods (used for multiple subassemblies), which cannot be returned to their original state without extensive processing. The assignment of components as purchased parts is based on the typical benchmarks of industrial large-scale production.

A pouch cell is composed of several individual components. Inside the outer pouch foil is the cell stack, which is saturated in electrolyte. Electrical energy is conducted through the terminals. The surface interfaces of the terminals are typically secured with covering tape to prevent tearing of the pouch foil. The pouch foil is the outermost layer of the battery cell. It is typically composed of a laminated film such as polyethylene or polypropylene. The pouch foil serves as a physical barrier to prevent the electrolyte from leaking out and to protect the internal components from external damage. Electrolyte is a liquid or gel that conducts ions, allowing the battery to charge and discharge. It typically contains lithium salts such as lithium hexafluorophosphate (LiPF6) or lithium tetrafluoroborate (LiBF4) dissolved in a solvent such as ethylene carbonate or propylene carbonate. The terminals are metal tabs that protrude from the pouch, allowing the battery to be connected to an external circuit. They are typically made of materials such as nickel, aluminum, or copper, which are able to withstand the currents and temperatures generated during the battery's operation. The terminals are attached to the cell stack (secured by covering tape), which enables the flow of electrons [27,28].

The cell stack itself consists of alternating layers of anode and cathode sheets, each separated by a separator. The separator is lastly wrapped around the cell stack and secured

with a fixing strip. The separator is a thin sheet of material such as polyethylene or polypropylene, which is placed between the anode and cathode to prevent short-circuiting. It acts as a physical barrier, allowing ions to migrate between the anode and cathode while preventing direct contact between the electrodes [29,30].

Both electrodes, anode and cathode, are composed of a thin metal sheet and coating material. The metal foil is typically made of aluminum for the cathode and copper for the anode. The coating material contains all the dry powders that make up the chemistry of the battery cell, initially mixed with a solvent. The key ingredients are the active material, binders, conductive material such as carbon black, and other additives [29,30].

*3.2. Process Chain for Battery Cell Production*

In the following, the process chain for battery cell production is outlined. This paper presents the production process for a battery pouch cell (see Figure 4). Compared to the other formats, round and prismatic cells, there may be process changes, especially in the assembly. Battery cell production can be divided into electrode manufacturing, cell assembly, and cell finishing.

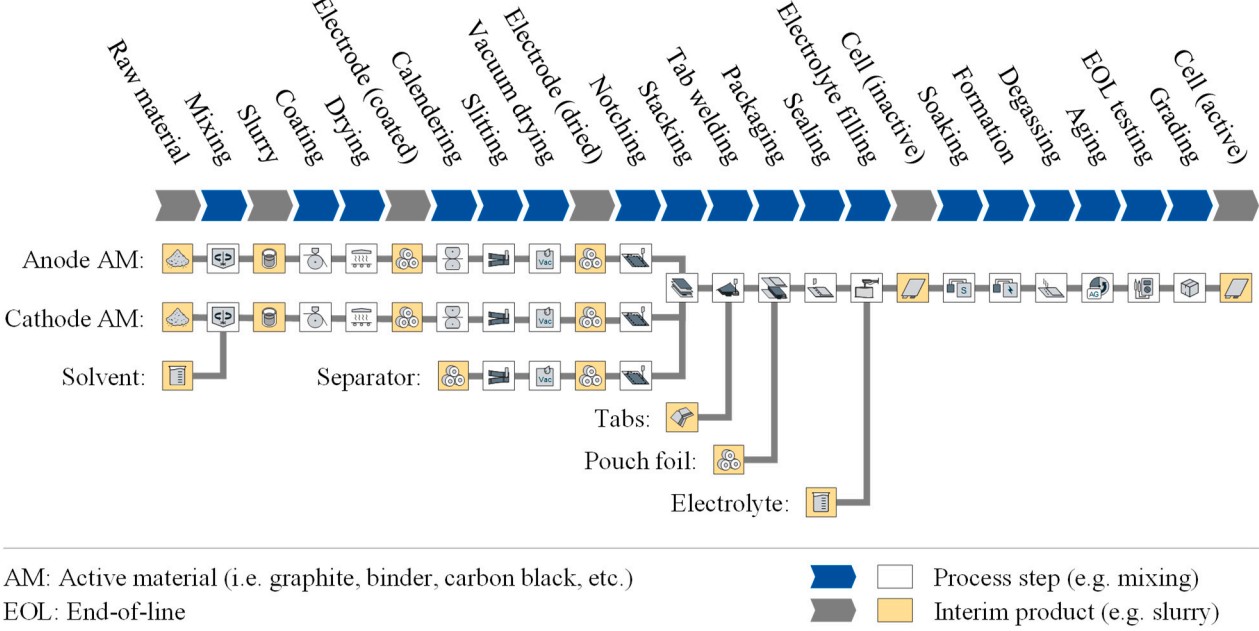

**Figure 4.** Generic process chain of a lithium-ion pouch battery cell with interim products.

Electrode manufacturing for pouch cells is a multi-step process that begins with mixing the active material, conductive agent, binders, and solvents to form a slurry. The slurry is made using purchased materials such as lithium cobalt oxide for cathode and graphite for anode; these materials are the active materials for the electrode. Once the slurry is prepared, it is coated onto a substrate, such as aluminum or copper foil. The coated substrate is then dried to remove any solvents and to solidify the active material. After the substrate is dried, it is passed through a calendering machine, which applies pressure to the substrate to control the thickness of the electrode and to define its porosity. After calendering, the substrate is then passed through a slitting machine, which cuts the substrate coils from the previous roll-to-roll process into smaller coils with the desired size. The final step in the electrode manufacturing process is vacuum drying, where the electrodes are dried to remove any residual moisture [29–31].

The cell assembly process of a battery pouch cell starts with notching single electrode sheets, which are cut into the desired shape and size. These electrodes are then stacked in an alternating order, with a separator film between the anode and cathode. After the electrodes are stacked, the next step is tab welding. During tab welding, metal tabs are

attached to the electrodes (in particular, the substrate film) to form the contact poles of the cell. This stacked assembly (cell stack) is then placed into a deep drawn pouch foil, which acts as the cell's housing. After the cell stack is placed into the pouch, the pouch cell is partly sealed and ready for electrolyte filling. The electrolyte is injected into the pouch through a small opening that is subsequently sealed to prevent leakage [29–31].

During cell finishing, the initial charging and discharging of the cell activates the electrodes and electrolyte, and establishes a stable quality and performance of the battery cell. The process begins with soaking to ensure that the electrolyte is completely soaked into the material and to create a stable interface. During formation, the initial charging and discharging of the cell takes place to activate the initial electrochemical reactions. During cell formation, the cell is cycled multiple times to improve its performance and stability. The next step is the aging process, where the cell is monitored regarding its electrical properties and to identify potential defects. This process can last anywhere from a few hours to several weeks. After aging, the cell is degassed to remove any dissolved gases that can negatively impact the performance and safety of the cell. In EOL testing, the battery cell is tested to ensure that it meets the desired performance specifications, such as capacity, voltage, internal resistance, self-discharge rate, temperature, and safety. Based on the EOL testing, the cells are graded into different categories based on their performance. Finally, the cell is packaged in a protective material to prevent damage during transport and is shipped to the customer [29–31].

*3.3. Process and Quality Parameters for Interim Products*

After discussion of the product structure for the lithium-ion pouch battery cell in Section 3.1 and the underlying process chain in Section 3.2, the identification, compilation, and assignment of the relevant parameters, which are included or recorded during the production of the battery cell, follows. For this purpose, an extensive study was carried out in which 150 sources were examined with regard to relevant parameters in battery production.

In total, 209 parameters were identified and allocated to the individual production steps along the reference process chain. A comprehensive overview of the identified parameters can be found in Table A1 in the Appendix A. This list forms the basis for the following considerations for an in-depth understanding of the mapping of parameters. However, the completeness of the list cannot be guaranteed, as some parameters are strongly dependent on the product design and choice of production technologies. Figure 5 shows their relative distribution and allocation of the identified parameters to the interim products. The calculation of the percentage values for the distribution of the parameters for each process step follows Equation (1), where $n_{process}$ is the number of parameters in each process step and $n_{total}$ is the total number of parameters. The values for the interim products result from the sum of parameters from all previous process steps.

$$x_{process} = \frac{n_{process}}{n_{total}} \tag{1}$$

It can be seen that about half (51.7%) of the total parameters are determined in electrode manufacturing. One reason for that might be that the early steps of the process chain place very high demands on quality, since any undetected defect is propagated through the entire following process chain and thus causes high scrap quantities [32]. These quality requirements can only be met by correspondingly extensive process monitoring, which results in the relatively large number of parameters. As mentioned in Section 3.2, the reference process chain consists of 18 individual production steps. After the first process step (mixing) 23% of the total parameters are already recorded (see Figure 5). This is primarily due to the high number of quality characteristics that are determined for the raw materials of the slurry, which is the case because the quality of the incoming goods greatly impacts the quality of outgoing products (battery cells).

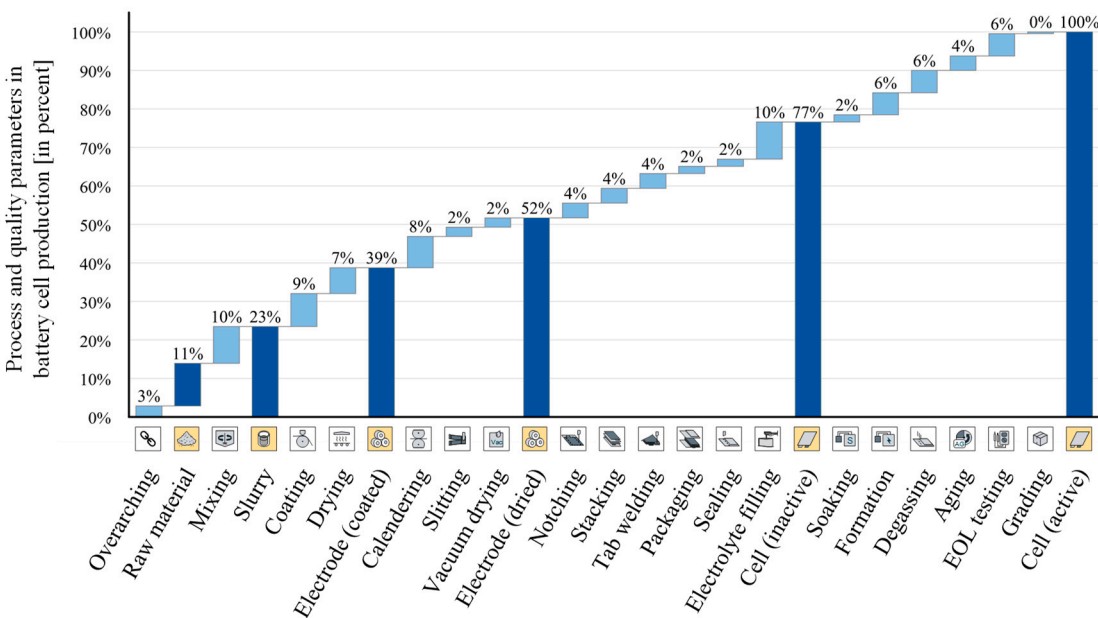

**Figure 5.** Relative distribution of process and quality parameters along the process chain.

Cell assembly and cell finishing share roughly the same number of collected parameters (24.9% and 23.4%). This coincides with the fact that these two parts of the process chain consist of six process steps each. In cell assembly, electrolyte filling and soaking take up the largest share of the parameters (in total, 11.5%). One reason for this is the fact that the cell is also sealed during electrolyte filling. Thus, two process steps are united here. In addition, soaking is a time-intensive process step. In order to be able to comply with the low cycle times required, an increased effort is presumably made here on the equipment side, which is reflected in the number of parameters.

In cell finishing, the number of parameters per process step is quite evenly distributed, with about 6% each of the total parameters. This is since the process steps of formation, aging, and EOL testing have some similarities. In each of the steps, the cell is charged/discharged, or the electrochemical properties are tested. This can be seen, for example, in the fact that the cell voltage or state of charge are measured in each of the steps. All these steps can thus also be carried out in the same system.

Some parameters, such as ambient conditions, are recorded or set in almost all process steps. In this analysis, they are therefore assigned once to the *overarching* category and not listed individually for each process. However, they might be more relevant in some process steps than others. For example, electrode manufacturing places high demands on particle contamination, whereas in cell assembly the humidity of the process environment plays a predominant role. Even though the number of parameters listed in this category is rather small, this does not reflect their relevance, especially regarding production costs.

### 3.4. Categorization and Mapping of Product and Production Information

In Section 3.3, an analysis of the process and quality parameters along the battery cell manufacturing chain was carried out. In order to be able to better structure a more detailed analysis of the parameters according to their origin along the manufacturing chain, this chapter provides a more in-depth categorization of the parameters considered. Such a categorization also provides an important basis for the development of the framework and an information model for the digital product twin, which is carried out in Section 4. Thus, the terms process and quality parameters are further specified. Consequently, a common terminology and definition of the product and production information is defined, which also will be part of the digital twin framework:

- Product feature: all properties and characteristics of intermediate products and the final product (cell) that can be measured or are given by the supplier, such as slurry viscosity, coating thickness of the electrode, or internal resistance of the final cell.
- Process parameter: all parameters that can be set directly on the respective process, such as mixer speed, web speed, or welding frequency.
- Equipment feature: parameters that cannot be changed at short notice in the process and are defined by the design of the equipment or its tools. Slot die width, drying line length, and calender roller diameter are examples of these kind of parameters.
- Ambient parameter: parameters that describe the conditions prevailing in the production process (humidity, temperature, etc.) and cannot be set directly on the respective machine, but are ensured by the room conditioning systems.

If the parameters from Table A1 in the Appendix A are categorized according to the definitions presented, we obtain predominantly product features (46.9%) and process parameters (44%). Equipment parameters (7.2%) and ambient parameters (1.9%), on the other hand, represent a small proportion of the list.

According to the distribution of product features in Figure 6, the majority of this parameter type is concentrated in electrode manufacturing. Accordingly, the indication given in Section 3.3 that the quality of the raw materials and intermediates plays an important role in electrode production can be reinforced.

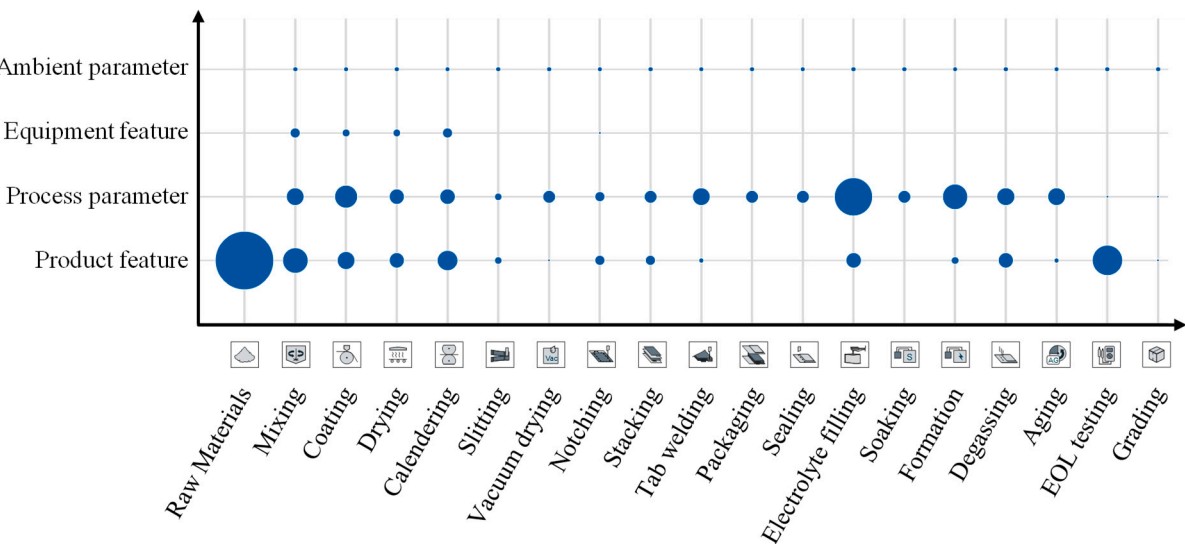

**Figure 6.** Distribution of the different parameter categories along the battery cell production chain.

Furthermore, the majority of the process parameters can also be assigned to electrolyte filling and soaking. This also confirms the assumption made in the previous chapter that many manipulated variables are present in these process steps. Similarly, the assumptions regarding the lack of process understanding can also be confirmed in this case.

Furthermore, it is noticeable that no process parameters are available for raw materials, EOL testing, and grading. In the case of raw materials, this can be explained by the lack of additional processing steps of these purchased parts. EOL testing and grading, on the other hand, are pure quality inspection and sorting processes that do not require any variable process settings.

Considering product features and process parameters along the entire process chain, it can be concluded that there is a close correlation between these two types of parameters. As an example, the set mixer speed during the mixing process step is a process parameter with a direct influence on the viscosity of the intermediate product electrode slurry, which represents a corresponding product feature [33]. Furthermore, the coating web speed influences the wet coating thickness of the electrode [34]. However, concrete evidence of

such cause–effect relationships cannot be determined from Figure 6. For this, concrete production data sets in combination with an evaluation within the framework of digital product twinning are required in order to be able to prove such cause–effect relationships.

The identified equipment features are almost exclusively concentrated in the area of electrode production, as can be seen in Figure 6. The largest share is accounted for by equipment settings in the course of the mixing, coating, and drying process. The cause for this could be heterogeneous production processes. For example, mixing is a process engineering production process, whereas slitting (without laser process) is a mechanical production process. Another possible reason is the amount of individual production equipment used in electrode manufacturing in comparison to a largely standardized assembly process. For example, the used mixer type (e.g., rotary, planetary, or extrusion mixer) can vary, and the number of temperature zones in the dryer depend on the line configuration [31].

A total of four parameters describing the ambient conditions of battery cell production were identified. Even though these parameters are classified as overarching in Figure 6, the same relevance of each of these parameters for each manufacturing step cannot be deduced from this. For example, humidity in the process environment has a very negative effect on the properties of the intermediate products. In the area of cell assembly, an excessively high moisture content in the production environment can lead to a short-circuit in the battery cell, which is why these process steps are usually carried out in dry rooms. In cell finishing, on the other hand, the cell is generally closed, which is why there are no special requirements for humidity in the process environment [31,35]. Thus, the relevance of these environmental parameters, which are shown in Figure 6 for all manufacturing processes, depends specifically on the individual manufacturing processes. A specific assessment can also only be made by means of concrete analyses in connection with the digital product twin.

In summary, it can be derived primarily that various process parameters, which can be set directly in the process, influence the quality characteristics of the (intermediate) products. Nevertheless, equipment and ambient parameters also play a relevant role, which is why all these different types of parameters should be included in the framework of the digital twin. In order to be able to make a profound statement regarding the complex interdependencies between the parameters considered in this chapter, a dedicated correlation analysis needs to be carried out by using suitable production data sets and data analysis methods in combination with the digital product twin.

## 4. Digital Product Twin for Battery Cell Production

In the previous chapters, relevant process variables in battery cell production were identified. In the following chapter, a concept for a digital product twin for battery cell production is presented based on these variables. The first step is to present the individual components of the twin. Then, the implementation and structuring of the data in the twin will be explained. Finally, information interfaces and interactions of the twin with external entities are explained.

### 4.1. Framework of a Digital Battery Product Twin

As previously described, the production of lithium-ion battery cells is still hindered by a high scrap rate due to the lack of knowledge surrounding the relationship between production and product quality [20]. In particular, the causes of quality outliers in mass production can often be difficult or even impossible to identify due to the large number of potential influences and disruptions. To address this, a promising approach is to capture, analyze, and compare as much data as possible from production to uncover the root causes of scrap and quality variances.

However, the causes of quality variances can be highly diverse and based on interactions between different parameters, as described in Section 3.4, which is why data from a large number of cells must be compared and analyzed. The amount of data increases significantly with the number of cells. To get an idea of the amount of data, a simple calcu-

lation can be made. The GigaFactory of Tesla and Panasonic in Nevada had a production capacity of 30 GWh/a in 2019. A common 21,700 cell has an energy content of about 18 Wh. This means that about 1.7 billion cells were produced at the Nevada production facility in a single year. Furthermore, the data must be structured to allow semantic and therefore human analysis.

The framework for the semantic structure of the digital product twin is derived from the product structure in the form of the bill of material, which was presented in Section 3.1. Figure 7 shows an exemplary representation of this incremental structure of the battery cell. In this structure, the product battery cell can be broken down to the intermediate products that led to its production (e.g., the electrode slurry or sheets) and the used raw materials and purchased parts (e.g., graphite, binders, or pouch foil). Furthermore, we differentiate between the assembled cell after the process stage of cell assembly and the activated cell after formation. The assembled cell is again divided into its individual parts or intermediate products, for instance individual sheets. Such a subdivision makes it possible to store characteristic data of the entire system (e.g., battery cell) as well as individual components' data in a structured way. Various sources can serve as data input for the digital battery twin, such as material properties, data sheets of components and equipment features, as well as production data and measurements. Material data can come from suppliers, for example. Particularly relevant material properties should be checked as part of an incoming goods inspection, which generates further data. The inspection of incoming materials is particularly relevant, as these have a major influence on the quality of the final product [36]. Equipment features are particularly relevant to ensure the comparability of cells from different production lines. For example, the type of equipment has an influence on the interpretation of the data. For example, the parameters mixing time and mixing tool speed are strongly dependent on the mixer type. Physical effects such as electrochemical, thermal, and electrical behavior of the active battery cell can then be modeled and simulated in a digital product twin based on that structure and the respective input data.

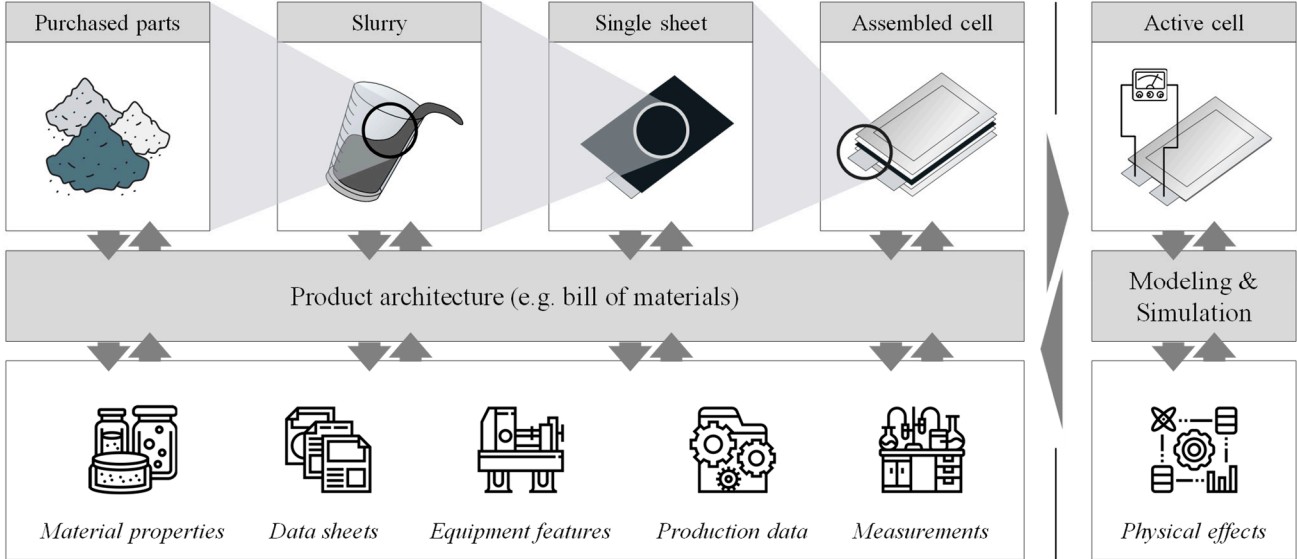

**Figure 7.** Framework for a semantic structure of a digital battery cell twin.

The presented framework contributes a more detailed data basis, which makes it possible to discover new interrelationships along the entire production chain and to better understand known correlations and influences through the means of process simulation and data analytics. This applies not only to the final product, the active battery cell, but also to the intermediate products and raw materials used. For example, measured quality properties after the mixing process of the electrode slurry could be used to adjust the process parameters in the subsequent coating process. In addition, the extended use of

artificial intelligence and machine learning also could provide opportunities to implement predictive quality and process control applications based on this structure.

The implementation of the framework should provide a suitable data and information interface between the relevant entities along the battery cell production chain, the intermediate as well as final products and represents the foundation for deeper analyses, and a better understanding of the complex interrelationships, support for decision making, or future predictive models. Eventually, this detailed structure also can be variably adapted to different cell formats and chemistries and also cover different production processes (with different manufacturing procedures and equipment) and quality inspections along the battery cell production chain.

### 4.2. Implementation and Data Structuring

In order to implement a digital twin, a corresponding information model was first developed that depicts the detailed collection of relevant data in a format that can be understood by experts and operators. A solution for implementing the presented information model is currently being developed based on Django, a Python-based framework. A section of this information model is shown in Figure 8. This is based on the scheme already presented in Section 4.1. Here, the cell is represented as a subassembly after cell assembly. Only after formation and after the recording of corresponding process data (see Section 3) is the battery cell considered to be the final product. The subassembly in turn consists of intermediate products (electrodes) and purchased parts (pouch foil, separator, etc.). The intermediate product electrode again consists of interim products and purchased parts (electrode made of slurry and substrate foil). In addition, each interim product has its own specific process parameters and product features as well as, under certain circumstances, equipment types which are also stored in defined memory locations. The data defined in this information model have already been explained in Section 3. Each subcomponent is also assigned a unique ID number, as is each purchased part. This makes direct access as well as a direct comparison possible also from components and intermediate products.

Through this structuring, each cell is broken down to all elementary components which it consists of. In addition, all influences (production parameters, product features, equipment parameters, and ambient parameters) that occurred during the manufacturing process of the individual cell are documented. This creates comparability of individual cells or batches, which in turn makes it possible to find the cause of quality defects and increased reject rates. Furthermore, in such an information model, sections of the total data set can be examined and accessed for further analyses. For example, it is possible to focus specifically on the intermediate product slurry. In addition, target values (such as a defined target value for the viscosity of the slurry) could be defined manually in this structure, with which the required process parameters could be determined and set for the machines.

In order to fill the developed information model with data, corresponding data mapping must take place, which enables the assignment of data sources to data targets. For this purpose, the following three steps need to be carried out: defining the relevant data, mapping the data, and transforming the data. First, the relevant data and its source, such as machine and measurement technology, need to be identified. These data are then located in a database. Next, the source fields need to be assigned to target fields, determining where the data would be stored. The data are then transformed, which involves converting data types, eliminating duplicates, and more. One of the challenges of mapping data is the presence of heterogeneous and different data sources, such as datasheets and measurements. With this approach, the challenges can be overcome and a scientifically sound method for mapping data in battery cell production is provided.

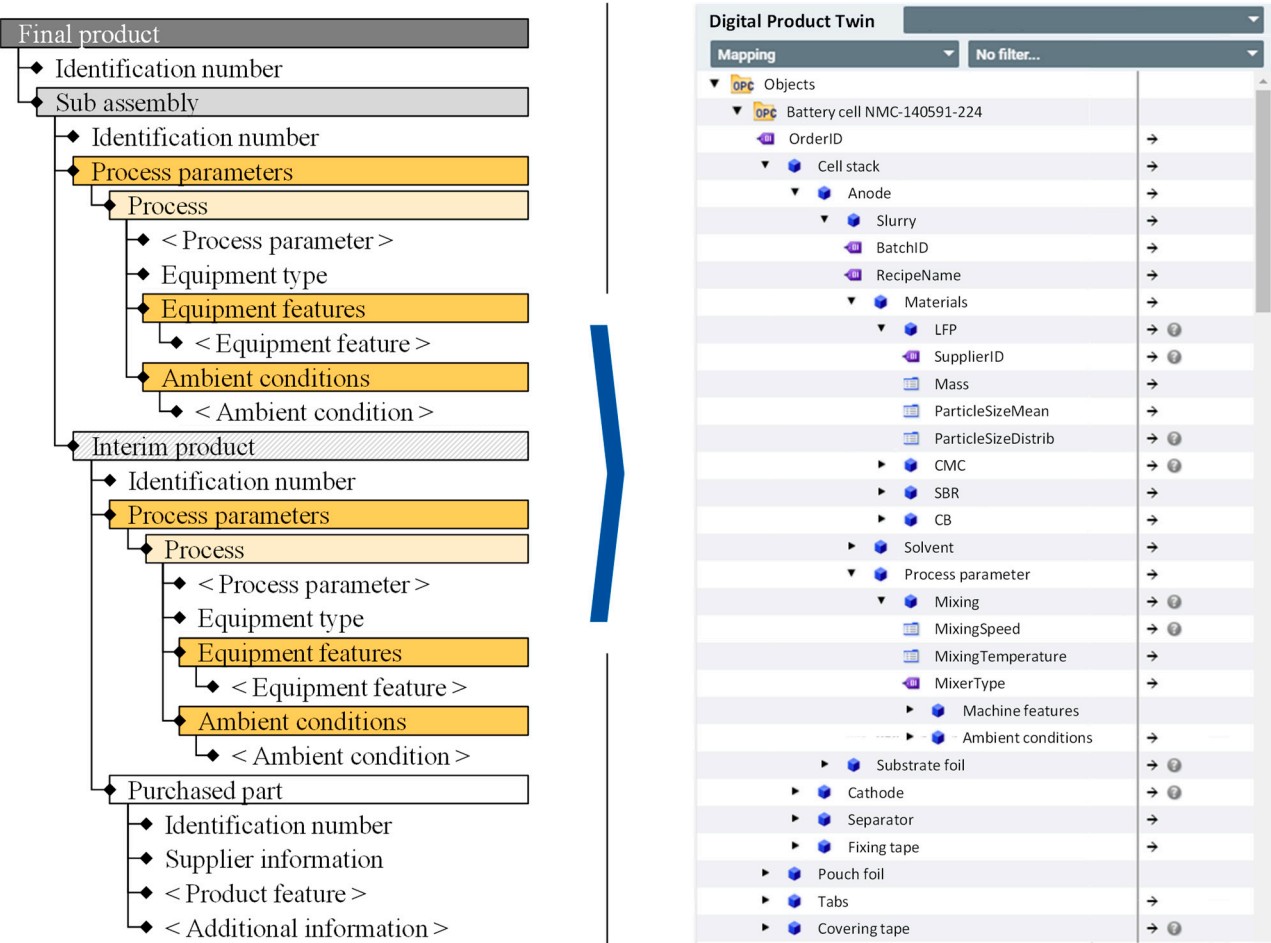

**Figure 8.** Section of an information model for a digital twin of a battery cell.

*4.3. Information Interfaces and Interactions*

Although the concept for a digital battery cell twin presented here has a focus on production, as this is where its characteristic data are generated, its use can go far beyond this. For example, the twin can be used in product development to iteratively develop an improved product, as shown in Figure 9. In addition to process and measurement data, design-relevant data such as geometric product specifications can also be stored in the product twin. Simulations can be carried out on the basis of the stored data. The results are then validated with the aid of the physical product (the battery cell). An example of this is that the real cell data from production form the input data for the thermal simulation of the cell at specific charge rates. The results can be checked on this cell specifically. If the simulation matches reality, optimization measures can be taken based on further simulations. For example, the definition of tighter specific manufacturing tolerances or the adaptation of the cell design.

To be able to implement all this, the product twin must offer appropriate interfaces. Information on material analyses, for example, can come from a LIMS (Laboratory Information and Management System). For thermal simulation, an interface (API) to simulation tools is necessary, and a CAD model must be stored for mechanical development. In order to implement the concept presented here, a very detailed traceability system is required. This must be able to guarantee traceability down to the electrode sheet level. Particularly with slurries, cell-specific traceability is difficult to achieve, since several slurries are often mixed in a buffer tank. If these hurdles can be overcome, the digital battery cell twin represents a suitable information interface between process information from the physical world and the digital simulation models.

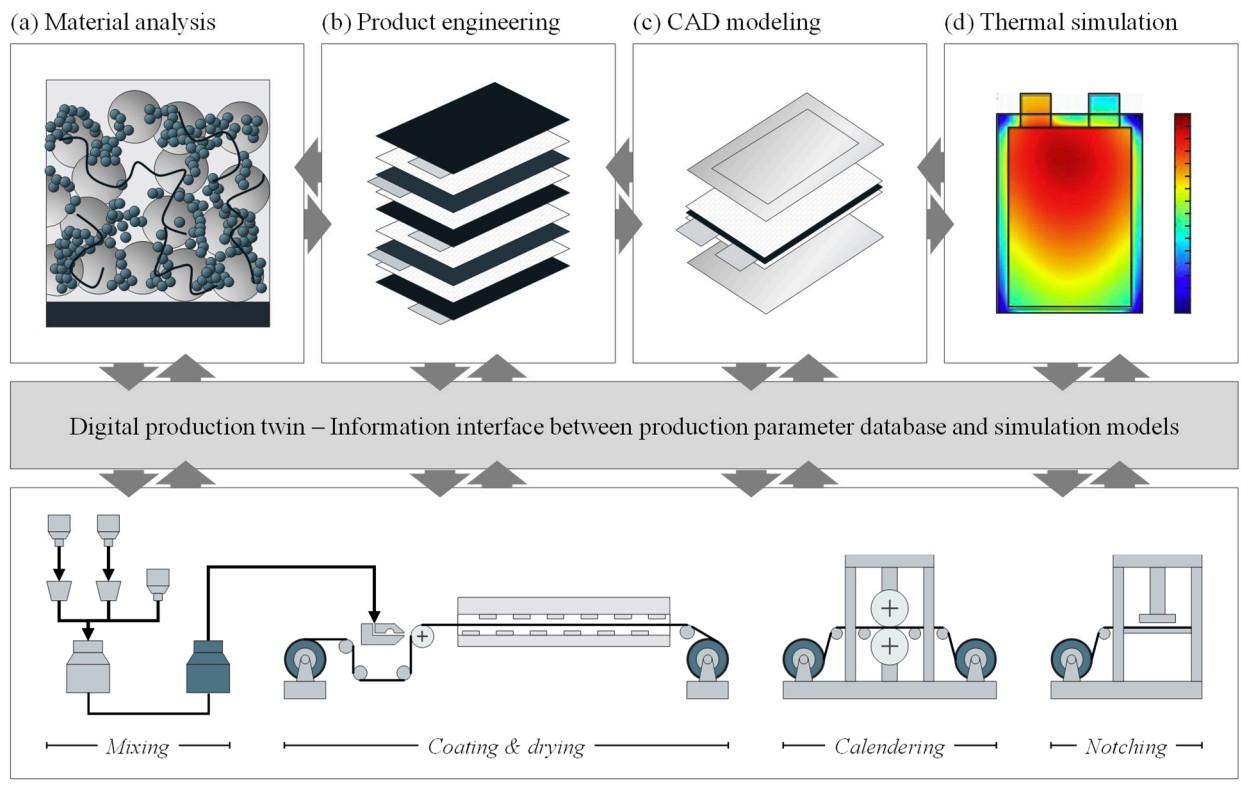

(a) Material analysis  (b) Product engineering  (c) CAD modeling  (d) Thermal simulation

Digital production twin – Information interface between production parameter database and simulation models

*Mixing*   *Coating & drying*   *Calendering*   *Notching*

(e) Process simulation

**Figure 9.** Interfaces and application fields for the digital twin.

## 5. Discussion

The need for a digital product twin of battery cells in battery cell production arises from the complexity of the manufacturing process and the need to optimize product quality, performance, and cost. A digital product twin is a virtual replica of a physical product, including its geometry, material properties, and manufacturing history. The digital product twin of a battery cell serves as an information interface between product characteristics and process parameter databases, enabling a variety of simulations and modeling by capturing and storing data on the manufacturing history of each individual battery cell, including its interim products and subassemblies along the process chain. Based on its product structure-related design, the digital product twin accommodates discrete, batch, and continuous production stages, for which the information along the process chain is continuously updated or supplemented in the product twin. The implementation of the digital product twin information model will enable manufacturers to systematically record the production process for each battery cell to analyze product quality and reliability from raw material correlations to individual process steps.

In this paper, a comprehensive overview of relevant data in battery production was presented. For this purpose, a comprehensive literature search was carried out. The result is a transparent list of parameters covering the entire process chain. Based on this, an analysis of the frequency of parameters per process step was carried out. Subsequently, the parameters were categorized in order to further increase their usability. Afterwards, it could be shown that the developed parameter list in connection with the categorizations can be used for the design of a digital twin. This was carried out by developing an information model which allows standardized storing of data from battery production. This can then further be used for simulations and process development.

In general, the number of papers dealing with the digital battery cell twin in the context of battery production is limited. Moreover, such a detailed parameter analysis in

battery cell production on which this is based does not yet exist in the literature, especially not in combination with such a categorization.

In many cases, the quality of a battery cell only becomes apparent during use. This is also the reason for the aging process step in cell production, which is merely a quality assurance measure. For this reason, further research could consist of developing a digital twin beyond the production boundaries (see Figure 10). By recording *field data* for specific battery cells and storing it in the digital cell twin, information can be obtained. By combining the field data with the corresponding production data for individual cells, the goal is to recognize cause–effect relationships. These can in turn be used to optimize production processes and thus result in increased product quality.

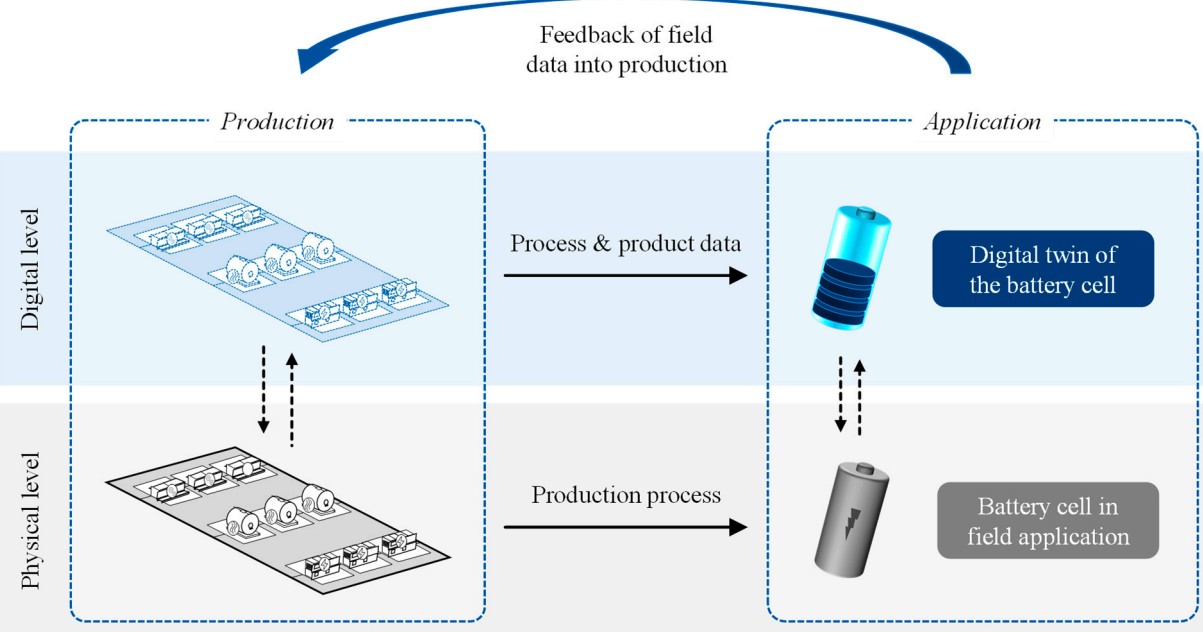

**Figure 10.** Potential expansion for a digital twin beyond the production boundaries.

**Author Contributions:** Conceptualization, H.C.; methodology, H.C.; formal analysis, H.C., R.L., R.L. (Ruiyan Li) and M.D.; investigation, H.C., R.L., R.L. (Ruiyan Li) and M.D.; data curation, R.L. (Ruiyan Li) and M.D.; writing—original draft preparation, H.C., R.L., R.L. (Ruiyan Li) and M.D.; writing—review and editing, B.D.; visualization, H.C.; supervision, H.C. and R.L.; project administration, H.H.H. and A.K.; funding acquisition, H.H.H. and A.K. All authors have read and agreed to the published version of the manuscript.

**Funding:** This research is based on the research results of the project "DataBatt" [03XP0323A]. The projects are funded by the German Federal Ministry of Education and Research. The authors of this publication thank the ministry and their partners for the funding.

**Data Availability Statement:** Not applicable.

**Conflicts of Interest:** The authors declare no conflict of interest.

## Appendix A

**Table A1.** Overview of battery cell production parameters for a pouch cell.

| Process | Parameter | Information | Unit | Source |
|---|---|---|---|---|
| Overarching | Atmosphere | Ambient parameter | - | [31] |
| | Humidity (dew point) | Ambient parameter | °C | [31] |
| | Clean room | Ambient parameter | ISO | [31] |
| | Ambient temperature | Ambient parameter | °C | [31,37,38] |
| | Process duration | Process parameter | min | [39] |
| | Power demand | Process parameter | kW | [39] |
| Raw material | Tap density (active material) | Product feature | $kg/m^3$ | [40,41] |
| | Purity (active material) | Product feature | % or ppm | [42] |
| | Humidity (active material) | Product feature | % | [42,43] |
| | Particle size distribution (active material) | Product feature | µm | [40,42,44,45] |
| | Particle shape (active material) | Product feature | - | [42] |
| | Specific surface (active material) | Product feature | $m^2/g$ | [40] |
| | Chemical composition (active material) | Product feature | - | [44] |
| | Chemical composition (electrolyte) | Product feature | % | [43] |
| | Tortuosity (separator) | Product feature | - | [46] |
| | Porosity (separator) | Product feature | % | [42,47] |
| | Thickness (separator) | Product feature | µm | [42,47] |
| | Puncture resistance (separator) | Product feature | J | [42] |
| | Temperature stability (separator) | Product feature | °C | [42] |
| | Yield stress (separator) | Product feature | MPa | [42] |
| | Yield strain (separator) | Product feature | % | [42] |
| | Max. stress (separator) | Product feature | MPa | [42] |
| | Nominal elongation at break (separator) | Product feature | % | [42] |
| | Polymer chain length (binder) | Product feature | - | [44] |
| | Polydispersity (binder) | Product feature | PI | [44] |
| | Purity (electrode foil) | Product feature | % | [42] |
| | Thickness (electrode foil) | Product feature | µm | [42,48] |
| | Surface roughness (electrode foil) | Product feature | µm | [49] |
| | Impurity (electrode foil) | Product feature | % | [42] |
| Electrode manufacturing—Mixing (dry) | Slurry formulation | Product feature | g or wt% | [40,44,45,48,50,51] |
| | Mixer type | Equipment feature | - | [44,45] |
| | Tank capacity | Equipment feature | L | [44] |
| | Mixing duration | Process parameter | min | [31,38,40,44,48,50] |
| | Mixing temperature | Process parameter | °C | [31,44,52] |
| | Mixing speed | Process parameter | RPM | [38,40,44,45,48] |
| | Agglomerate size | Product feature | µm | [44,53] |
| Electrode manufacturing—Mixing (dispersing) | Mixer type | Equipment feature | - | [44,45] |
| | Tank capacity | Equipment feature | L | [44] |
| | Mixing duration | Process parameter | min | [31,38,40,44,48,50] |
| | Mixing temperature | Process parameter | °C | [31,44,52] |
| | Mixing speed | Process parameter | RPM | [38,40,44,45,48] |
| | Viscosity | Product feature | mPas | [31,38,40,44,45,50–53] |
| | Agglomerate size | Product feature | µm | [31,44,45,52,53] |
| | Surface tension | Product feature | N/m | [44] |
| | Slurry density | Product feature | $g/cm^3$ | [44] |
| | Solids content of the slurry | Product feature | wt% | [38,44,45,51–54] |
| | Slurry purity | Product feature | % | [31,52] |
| | Diffusion coefficient of the active material | Product feature | $m^2/s$ | [47] |
| | Electrical conductivity of the slurries | Product feature | S/m | [47,55] |

**Table A1.** *Cont.*

| Process | Parameter | Information | Unit | Source |
|---|---|---|---|---|
| Electrode manufacturing—Coating | Slot die distance | Equipment feature | µm | [44,50,54] |
| | Slot die angle | Equipment feature | ° | [44] |
| | Web tension | Process parameter | N/mm$^2$ | [44,51] |
| | Foil folding (wrinkle) | Process parameter | µm | [51] |
| | Operating speed | Process parameter | m/min | [44,48,50] |
| | Pump flow rate | Process parameter | m$^3$/min | [44] |
| | Slot width | Equipment feature | mm | [31,44] |
| | Temperature of the coating material | Process parameter | °C | [44,52] |
| | Coating accuracy (mismatch) | Product feature | % | [31,52] |
| | Web edge | Process parameter | µm | [50,52] |
| | Coating thickness (wet) | Product feature | µm | [38,44,45,47,50,52,54,55] |
| | Coating accuracy (wet) | Process parameter | % | [31,52] |
| | Coating weight/weight per unit | Product feature | g/m$^2$ | [38,44,56] |
| | Shear rate (slot) | Process parameter | s$^{-1}$ | [44] |
| | Viscosity | Product feature | mPas | [44,50,51,53] |
| | Defects | Product feature | - | [31,44] |
| | Particle sizes of the coating | Product feature | µm | [38,44,47] |
| | Coating porosity | Product feature | % | [38,44,47,52,56] |
| Electrode manufacturing—Drying | Coating thickness (dry electrode, uncalendered) | Product feature | µm | [30,31,38,44,45,47,48,54,57] |
| | Drying line | Equipment feature | m | [31,44] |
| | Number of temperature zones | Equipment feature | - | [44,54] |
| | Temperature profile in the dryer zone | Process parameter | °C | [31,38,44,54] |
| | Web speed | Process parameter | m/min | [31,38,44,50] |
| | Air velocity | Process parameter | m/s | [44,56] |
| | Air nozzle spacing | Equipment feature | m | [44] |
| | Air volume flow | Process parameter | m$^3$/min | [38,44,56] |
| | Web tension | Process parameter | N/mm$^2$ | [31,44,51] |
| | Fractures in the material | Product feature | µm | [31,44,52] |
| | Residual humidity | Product feature | % | [38,52] |
| | Binder and conductivity additive migration | Product feature | % or ppm | [50–52] |
| | Adhesion/adhesive strength | Product feature | N/mm$^2$ | [38,44,45,49,51,52] |
| | Coating porosity | Product feature | % | [38,44,45,47,52,56] |
| Electrode manufacturing—Calendering | Foil folding (wrinkle) | Process parameter | µm | [51] |
| | Defects | Product feature | - | [44] |
| | Roller width | Equipment feature | m | [31] |
| | Roller surface roughness | Equipment feature | µm | [31] |
| | Roller concentricity | Equipment feature | µm | [31] |
| | Roller diameter | Equipment feature | mm | [31] |
| | Line pressure | Process parameter | N/mm | [31,38,44,50] |
| | Temperature control of the roller | Process parameter | °C | [31,38,44] |
| | Roller drive velocity | Process parameter | min$^{-1}$ | [44] |
| | Gap size | Process parameter | µm | [44] |
| | Adhesion/adhesive strength | Product feature | N/mm$^2$ | [31,38,44] |
| | Surface roughness | Product feature | µm | [31,38] |
| | Tortuosity | Product feature | - | [30,46,55] |
| | Pore size distribution | Product feature | (µm) | [30,44] |
| | Coating porosity | Product feature | % | [31,38,44,47,50,51,55,56,58] |
| | Coating weight/weight per unit | Product feature | g | [44,52] |
| | Coating thickness (calendered electrode) | Product feature | µm | [38,44,47,50,52,55] |
| Electrode manufacturing—Slitting | Cutting speed | Process parameter | m/min | [31] |
| | Cutting edge geometries | Process parameter | µm | [31,38,50,52] |
| | Foreign particles | Product feature | µm | [31,52,59] |
| | Microstructure deformation | Product feature | µm | [52] |
| | Burr height | Product feature | µm | [38] |

**Table A1.** *Cont.*

| Process | Parameter | Information | Unit | Source |
|---|---|---|---|---|
| Electrode manufacturing—Vacuum drying | Temperature | Process parameter | °C | [31,38] |
| | Drying time | Process parameter | h | [31,38,48,50] |
| | Vacuum pressure | Process parameter | mbar | [31,38,48] |
| | Humidity (dew point) | Process parameter | °C | [50] |
| | Residual humidity | Product feature | % or ppm | [31,38] |
| Cell assembly—Notching | Punching time | Process parameter | s | [31,60] |
| | Punching speed | Process parameter | s/sheet | [31,60] |
| | Wear resistance (tool life) | Equipment feature | - | [31,60] |
| | Electrode geometry | Product feature | mm | [31,60] |
| | Cutting accuracy | Process parameter | μm | [51,60] |
| | Electrode cutting height | Product feature | μm | [60] |
| | Cut size variation | Product feature | μm | [60] |
| | Electrode tortuosity | Product feature | - | [30,46,55] |
| Cell assembly—Stacking | Positioning accuracy of the electrode sheets | Process parameter | μm | [31,51,52,61] |
| | Suction pressure of the gripper | Process parameter | Pa | [38] |
| | Separator pre-tension | Process parameter | MPa | [62] |
| | Stacking accuracy | Product feature | μm | [31,43] |
| | Number of sheets | Product feature | - | [48] |
| | Foreign particle concentration | Product feature | $1/m^3$ | [52,59] |
| | Electrical charge | Product feature | C | [31,52] |
| | Clamping force of the hold-down | Process parameter | N | [38,43] |
| Cell assembly—Tab welding | Amplitude | Process parameter | μm | [63] |
| | Welding force | Process parameter | N | [43,63] |
| | Frequency | Process parameter | kHz | [31] |
| | Welding time | Process parameter | s | [63] |
| | Holding force of the cell tab contact | Process parameter | N | [52] |
| | Contact resistance of the cell tab | Product feature | S | [52] |
| | Optical inspection of the cell tab | Product feature | - | [31,52] |
| | Weld energy | Process parameter | kJ/cm | [63] |
| Cell assembly—Packaging | Hold-down force | Process parameter | N | [64] |
| | Stamp speed | Process parameter | Stroke/min | [64] |
| | Stamping force | Process parameter | N | [64] |
| | Temperature | Process parameter | °C | [64] |
| Cell assembly—Sealing | Sealing pressure | Process parameter | $N/mm^2$ | [31,43] |
| | Sealing temperature | Process parameter | °C | [31] |
| | Sealing duration | Process parameter | s | [48] |
| | Vacuum pressure | Process parameter | mbar | [48] |
| Cell assembly—Electrolyte filling (filling) | Volume flow | Process parameter | $m^3$ | [37,43] |
| | Electrolyte quantity | Process parameter | ml | [31,43,65–67] |
| | Number of filling cycles | Process parameter | - | [31,66,67] |
| | Vacuum pressure | Process parameter | mbar | [31,38,66] |
| | Vacuum time | Process parameter | s | [43] |
| | Filling duration | Process parameter | s | [51,66–68] |
| | Electrolyte temperature | Process parameter | °C | [37,38,52,67] |
| | Cell weight | Product feature | g | [48] |
| | Diffusion coefficient of the electrolyte | Product feature | $m^2/s$ | [55] |
| Cell assembly—Electrolyte filling (wetting) | Wetting duration | Process parameter | s | [37,43,66–68] |
| | Operating pressure | Process parameter | bar | [31,43,66–68] |
| | Degree of wetting/distribution of the electrolyte | Product feature | % | [31,38,66–68] |
| | Electrical insulation resistance | Product feature | Ω | [52] |
| Cell assembly (pouch)—Electrolyte filling (sealing under vacuum) | Sealing temperature | Process parameter | °C | [48] |
| | Vacuum pressure | Process parameter | $N/mm^2$ | [48] |
| | Sealing temperature | Process parameter | °C | [31] |
| | Sealing pressure | Process parameter | $N/mm^2$ | [31,37] |
| | Sealing duration | Process parameter | s | [48] |
| | Inspection of the sealing | Product feature | - | [52] |
| | Leakage | Product feature | - | [31,52] |

**Table A1.** *Cont.*

| Process | Parameter | Information | Unit | Source |
|---|---|---|---|---|
| Cell finishing—Soaking | Soaking time | Process parameter | h | [39,43,69] |
| | Temperature | Process parameter | °C | [39] |
| | Vacuum pressure | Process parameter | Mbar | [43] |
| | Vacuum time | Process parameter | h | [43] |
| Cell finishing—Formation | Contact resistances at the spring contacts | Process parameter | S | [31] |
| | Formation cycle duration | Process parameter | h | [38] |
| | Charging range (SOC) | Process parameter | % | [31,70] |
| | Charging voltage | Process parameter | V | [31,37] |
| | Charging current | Process parameter | A | [31,37] |
| | Charge and discharge cycles | Process parameter | Cycles | [31] |
| | Temperature | Process parameter | °C | [31,38] |
| | Compression pressure during formation | Process parameter | MPa | [31,37] |
| | Precharge duration | Process parameter | h | [31,38] |
| | Cell temperature | Product feature | °C | [37,52] |
| | Cycle efficiency | Product feature | - | [38,70] |
| | Discharge capacity | Product feature | Ah | [51,70] |
| Cell finishing—Degassing (piercing the gas bag and suction of gas) | Contact pressure on the cell | Process parameter | MPa | [31] |
| | Weight | Product feature | g | [37] |
| | Residual gas inside cell | Product feature | ml | [67,68] |
| Cell finishing (pouch)—Degassing (final sealing) | Sealing temperature | Process parameter | °C | [31] |
| | Sealing pressure | Process parameter | N/mm$^2$ | [31] |
| | Sealing duration | Process parameter | s | [48] |
| | Vacuum pressure | Process parameter | bar | [31] |
| | Vacuum time | Process parameter | s | [31] |
| | Seam width | Product feature | mm | [31] |
| | Heat input | Product feature | W | [31] |
| Cell finishing—Degassing (folding) | Leakage | Product feature | - | [31,38,52] |
| | Folding edge geometry | Product feature | µm | [31] |
| Cell finishing—Aging (HT-aging) | Aging duration | Process parameter | h | [31] |
| | Aging temperature | Process parameter | °C | [31] |
| | SOC | Process parameter | % | [30,31] |
| | Voltage loss rate | Product feature | % | [30] |
| Cell finishing—Aging (NT-aging) | Aging duration | Process parameter | days | [31] |
| | Aging temperature | Process parameter | °C | [31] |
| | SOC | Process parameter | % | [30,31] |
| | Voltage loss rate | Product feature | % | [30] |
| Cell finishing—EOL testing | SOC of the cell for shipping | Product feature | % | [20,30,71] |
| | Optical inspection | Product feature | - | [31,71] |
| | Electrical-dynamic behavior | Product feature | - | [31,52,71] |
| | Electrical internal resistance | Product feature | Ω | [31,52,71] |
| | Impedance | Product feature | Ω | [56,70] |
| | Voltage | Product feature | V | [30,71] |
| | Leakage | Product feature | - | [31,52] |
| | Foreign particle concentration | Product feature | 1/m$^3$ | [59] |
| | Thermal conductivity of the cell | Product feature | W/mK | [72] |
| | Heat capacity of the cell | Product feature | J/kg·K | [72] |
| | Cell temperature | Product feature | °C | [37,38] |
| | Cell weight | Product feature | g | [37,71] |
| Grading | Grade | Product feature | - | [31,71] |

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
