# Peer review of "Concept for Digital Product Twins in Battery Cell Production"

_wevj, doi:10.3390/wevj14040108_

Round 1

Reviewer 1 Report

The paper is well-writtena nd well organised, it covers most of the relevant research studies such as those covered by ARTISTIC and NEXTRODE project.

The organisation is also well decided and given the presentation and summary, I would be more than happy to recommend its publication in its current form

Author Response

Dear reviewer,

Thank you for taking the time to evaluate our work. We appreciate your comments and thank you again for your valuable feedback. 

Yours sincerely

Reviewer 2 Report

Thank you for inviting me to review the manuscript below:

Manuscript ID wevj-2280601

Title: Concept for Digital Product Twins in Battery Cell Production

This paper presents a methodical approach to the design and derivation for establishing a digital product twin for battery cells. The article contains no simulation model or research methodology, and a new scientific contribution has not been detected.  it looks like a report without any scientific added value. For that, I cannot propose publishing this article in MDPI WEVJ.

Author Response

Dear reviewer,

Thank you for taking the time to evaluate our work. We appreciate your comments and thank you again for your valuable feedback. We hope that we have addressed your concerns as much as possible in our revisions.

For the response to your comments see the attached document.

Yours sincerely

Reviewer 3 Report

Please incorporate the below points to improve the manuscript: 

1) Introduction needs to be strengthened focusing on current challenges to developing digital twins in battery cell production, approaches adopted so far, and the need for the present work.

2) Section 2.1 seems unnecessarily long explaining aspects of digital twins development. It should be shortened and more focus is to be given on battery production. 

3) Fig 2 seems quite obvious and generic. Please make it specific to the battery cell production where information flow and feedback loop, if any are to be included. 

4) Sections 3.1 and 3.2 are also well-known in the literature. What is new here?

5) Figure 5 shows their relative distribution and allocation of the identified parameters to the interim products. How the relative distribution was conducted. No Information on percent calculation. 

6) The framework for Digital Product Twin for Battery Cell Production in Section 4 seems weak considering the previous background and process mapping. Please explain what benefits can be obtained from this. 

7) Discussion is very limited. It should not summarize what you have done but rather the rationale behind it. Its better to explain with a case study how the developed digital twin is helpful in understanding battery cell production.   

Author Response

(The authors gave the same response as above.)

Round 2

Reviewer 2 Report

Thank you for inviting me to review the manuscript below:

Ms. Ref. No.:  APEN-D-20-07405

Title: Concept for Digital Product Twins in Battery Cell Production

Journal: WEVJ (ISSN 2032-6653)

This paper focuses on the concept of digital product twins in battery cell production. The paper needs minor revision.

1.      The novelty/originality shall be further justified by highlighting that the manuscript contains sufficient contributions to the new body of knowledge. The knowledge gap needs to be clearly addressed in the Introduction.

2.      In the text there are errors in English, that needs to be carefully read and corrected.

Author Response

Dear reviewer,

thank you for taking the time and for the valuable feedback.  We have considered your comments and made the necessary revisions.

Best regards

Reviewer 3 Report

Thank you for the updated manuscript. 

Author Response

Dear reviewer,

thank you for taking the time and your valuable feedback. 

Best regards
